# Three-Arm Robotic Lung Resection via the Open-Thoracotomy-View Approach Using Vertical Port Placement and Confronting Monitor Setting: Focusing on Segmentectomy

**DOI:** 10.3390/jpm12111771

**Published:** 2022-10-27

**Authors:** Noriaki Sakakura, Takeo Nakada, Yusuke Takahashi, Ayumi Suzuki, Shuichi Shinohara, Hiroaki Kuroda

**Affiliations:** Department of Thoracic Surgery, Aichi Cancer Center Hospital, 1-1 Kanokoden, Chikusa-ku, Nagoya 464-8681, Japan

**Keywords:** robotic lung resection, open-thoracotomy-view approach, vertical port placement, confronting monitors, segmentectomy, lobectomy

## Abstract

To perform robotic lung resections with views similar to those in thoracotomy, we devised a vertical port placement and confronting upside-down monitor setting: the three-arm, robotic “open-thoracotomy-view approach (OTVA)”. We described the robotic OTVA experiences focusing on segmentectomy and its technical aspects. We retrospectively reviewed 114 consecutive patients who underwent robotic lung resections (76 lobectomies and 38 segmentectomies) with OTVA using the da Vinci Xi Surgical System between February 2019 and June 2022. To identify segmental boundaries, we administered indocyanine green intravenously and used the robotic fluorescence imaging system (Firefly). In all procedures, cranial-side intrathoracic structures, which are often hidden in the conventional look-up-view method, were well visualized. The mean durations of surgery and console operation were 195 and 140 min, respectively, and 225 and 173 min, for segmentectomy and lobectomy, respectively. In segmentectomy, console operation was significantly shorter (approximately 30 min, *p* < 0.001) and two more staplers (8.2 ± 2.3) were used compared with lobectomy (6.6 ± 2.6, *p* = 0.003). In both groups, median postoperative durations of chest tube placement and hospitalization were 0 and 3 days, respectively. This three-arm robotic OTVA setting offers natural thoracotomy views and can be an alternative for segmentectomy and lobectomy.

## 1. Introduction

We recently reported the three-arm, robotic “open-thoracotomy-view approach (OTVA)” using vertical port placement and confronting upside-down (CUD) monitor setting to perform robotic lung resections (RLRs) [1,2]. In our practice, we routinely performed open-thoracotomy surgery (OTS) using the vertical muscle sparing/splitting thoracotomy (VMST) [3], with the operating surgeon standing on the right side of the patient regardless of the side to be operated on. During a video-assisted thoracoscopic surgery (VATS), the operating surgeon also stands on the right side of the patient and uses the CUD monitor setting [4]. This allows the operating and assisting surgeons to have the same surgical view in our OTS and VATS. To maintain this consistency even in RLRs, we devised our three-arm robotic OTVA. Currently, robotic lung segmentectomies have been widely performed [5,6,7,8,9,10,11,12]. Given that surgical views and OTVA settings are different from those in the well-established worldwide conventional look-up-view approach [13,14,15,16,17,18,19,20], some considerations are needed for this method. Herein, we discuss the technical aspects of the three-arm robotic OTVA focusing on segmentectomy.

## 2. Patients and Methods

### 2.1. Patients

Insurance coverage for robot-assisted thoracoscopic surgery (RATS) was initiated in April 2018 in Japan. In February 2019, we began to perform RATS at full scale, totaling 145 surgeries until June 2022. We retrospectively reviewed 114 consecutive patients who underwent major RLRs (lobectomy, *n* = 76; segmentectomy, *n* = 38) using the three-arm OTVA with the CUD monitor setting. The baseline characteristics of the patients are shown in Table 1. The institutional review board of the Aichi Cancer Center Hospital approved the study (#2021-0-223). Each patient provided informed consent for the use of their clinical data.

### 2.2. Surgical Indications and Assessment

The procedures were primarily performed for clinical stage I primary lung cancer and lesions strongly suspected to be early stage lung cancer based on the eighth tumor–node–metastasis classification system, and other resectable lung tumors. Preoperatively, the lesion and the lung, including pulmonary vessels, bronchi, and fissures, were thoroughly assessed using axial, sagittal, and coronal images of high-resolution computed tomography (Aquilion Prime SP, Canon Medical Systems Corp., Otawara, Japan) and its three-dimensional reconstruction (SYNAPSE VINCENT, FUJIFILM Corp., Tokyo, Japan). As small lesions with an almost ground-glass appearance have a small pathological invasive size based on preoperative high-resolution computed tomography [21], mediastinal lymph node dissection was omitted for such less invasive lesions. Segmentectomy was considered for such less invasive lesions, multiple lesions, or metastatic lung tumors. In all patients, common respiratory function test, echocardiography, treadmill exercise test, and blood gas analysis were performed to evaluate their cardiopulmonary status. All surgical procedures for each patient were decided after our department conferences. For pathological staging, in cases where mediastinal lymph node dissection was omitted, possible hilar lymph nodes were evaluated and the pN status and postoperative disease stage were determined.

### 2.3. Three-Arm Robotic OTVA Setting

#### 2.3.1. System Setting

The procedure of our three-arm robotic OTVA was reported previously in detail [1,2]. Patients were placed in conventional right or left lateral decubitus position and managed by general anesthesia and double-lumen intubation.

The da Vinci Xi^®^ Surgical System (Intuitive Surgical Inc., Sunnyvale, CA, USA) was used. The patient cart was always rolled in at an 45°–60° angle from the left cranial side of the patient, regardless of the side to be operated on. In the three-arm setting, arm 1 (unused arm) was pushed toward the cranial side of the patient (anesthesiologist’s side), arm 2 was positioned at the cranial side of the patient for the left hand of the console surgeon, arm 3 was used for the 30°-robotic endoscopy, and arm 4 was positioned on the caudal side of the patient for the right hand of the console surgeon.

#### 2.3.2. Port Placements

Robotic ports were placed vertically along the posterior (for right-side surgeries) or anterior (for left-side surgeries) axillary line (Figure 1, top). For example, in the right upper lobe, the 8 mm, 8 mm, and 12 mm robotic ports were located at the third, fifth, and seventh intercostal spaces, respectively, and an assistant port (Alnote Lapsingle, AL-LS-51-1318, Alfresa Pharma Corporation, Osaka, Japan) was placed on the fifth or sixth intercostal space. This setup was described as “3/5/7/A5 or A6” (Figure 1, top, R). As similarly described, following setups were used: “3/5/8/A6 or A7” or “4/6/8/A6 or A7” for the right middle lobe, “4/6/8/A6 or A7” for the right lower lobe, “3/5/7/A8 or A9” or “2/4/6/A7 or A8” for the left upper lobe (Figure 1, top, L), and “3/5/8/A9 or A10” or “4/6/8/A9 or A10” for the left lower lobe. In these setups, three robotic ports were placed around the VMST incision line [3]. An insufflation system (AirSeal^®^ System, ConMed Corporation, Utica, NY, USA) was used to maintain a stable intrathoracic positive pressure of 5–10 mmHg.

#### 2.3.3. Assistants and CUD Monitor Setting

Two confronting monitors and two assistants were positioned on each side of the patient (Figure 1, bottom). Regardless of the side to be operated on, assistant A standing on the patient’s right side (i.e., the patient’s dorsal side for the right-lung surgery or ventral side for the left-lung surgery) was mainly responsible for the docking/undocking procedures and the exchange of robotic instruments. Assistant B standing on the patient’s left side (i.e., the patient’s ventral side for the right-lung surgery or dorsal side for the left-lung surgery) directly assisted with the surgery (i.e., retracting lungs and other intrathoracic structures, suctioning blood, and firing nonrobotic staplers). The left-side monitor (set up for assistant A) showed the same image as on the surgeon console. The right-side monitor (set up for assistant B) projected the upside-down image of the surgeon console view. These settings enabled the console surgeon and the two assistants to naturally acquire the same views as in our OTS and VATS procedures. In a previous report, these setups of port placements were described in detail and the roles of two assistants and confronting monitors were visually demonstrated using a video [1].

#### 2.3.4. Selection of Instruments

In selecting Endowrist^®^ instruments, we preferred using the combination of fenestrated bipolar forceps (arm 2) and monopolar curved scissors (arm 4) for most dissecting maneuvers. The reason for selecting these instruments was that the primary surgeon (NS) preferred holding the long round point forceps in the left hand and the Metzenbaum scissors or Mayo scissors in the right hand during OTS. The monopolar curved scissors can cut tissues finely, although the dissecting speed is slower compared to the common Maryland bipolar forceps. In addition to these instruments, we frequently used Vessel Sealer Extend^®^ and medium-large clip appliers for lymph node dissection. A newer sealing device, SynchroSeal^®^, is not yet available at our institution. Robotic or nonrobotic staplers were used at the surgeon’s discretion, with a 30 mm stapler for the vessels and a 45 mm stapler for the lung parenchyma and bronchus.

### 2.4. Segmentectomy and Postoperative Management

For segmentectomies, after resecting the segment-specific pulmonary arteries, veins, and bronchi, we diluted 25 mg of indocyanine green with 10 mL of sterile water and administered 5 mg (2 mL) of the solution intravenously. The segmental boundaries were then identified using a fluorescence imaging system (Firefly^®^) and were marked using robotic energy devices (fenestrated bipolar forceps or monopolar curved scissors) and pyoktanin blue solutions.

Intraoperative air leaks were evaluated with the AirSeal stopped and the chest cavity filled with distilled water using the SuctionIrrigator^®^ to inflate the lungs. Fibrin glue (BOLHEAL^®^, KM Biologics Co., Ltd., Kumamoto, Japan) and polyglycolic acid sheet (NEOVEIL sheet 0.15, Gunze Ltd., Kyoto, Japan) were used according to the degree of air leak. A 20-Fr chest tube (Argyle™ Trocar Catheter, Nippon Covidien Inc., Fukuroi, Japan) was placed at the camera port. The tube was connected to a suction system (COMPACT DRAIN UNIT, Sumitomo Bakelite Co., Tokyo, Japan, or Chest Drain Bag, Sumitomo Bakelite Co., Tokyo, Japan), and the suction was initially maintained at −5 cm H_2_O.

Approximately 2 h postoperatively, intake of rich-fat ice cream was initiated. Furthermore, 2 h later, if no air leak or chyle was observed, the drainage volume was less than 100 mL, and no abnormalities such as collapsed residual lungs on chest X-ray were observed, the chest tube was removed even on the same day of surgery (postoperative 0 day) based on our early postoperative mobilization protocol [22,23]. Chest X-ray was also performed the day after tube removal, when no abnormalities (residual lung collapse, atelectasis, pneumonia, or subcutaneous emphysema) were detected, and the patients were discharged home.

### 2.5. Statistical Analysis

The baseline characteristics of the patients and the surgical outcomes were compared between the lobectomy and segmentectomy groups using the chi-square test, Fisher’s exact test, Mann–Whitney U-test, or Student’s *t*-test, where appropriate. The JMP for Windows (version 10.0, SAS Institute, Cary, NC, USA) was used for all statistical analyses.

## 3. Results

### 3.1. Surgical Views

In this OTVA setting, cranially located intrathoracic structures, which are often hidden in the conventional look-up-view method, were well visualized and confirmed. Figure 2, Figure 3, Figure 4 and Figure 5 and Appendix A show the procedures of robotic OTVA segmentectomies: (1) right upper S1+2 segmentectomy, (2) left upper divisionectomy, (3) left upper S3 segmentectomy, and (4) left lower basal divisionectomy. The console surgeon and two assistants can obtain natural “bird-eye” views and perform the surgery as though they were performing OTS or confronting VATS.

### 3.2. Surgical Outcomes and Comparison between Lobectomy and Segmentectomy

Surgical outcomes are summarized in Table 2. The segmentectomy group tended to have an earlier pathological disease stage than the lobectomy group. Mean durations of surgery and console operation of the lobectomy group were 225 and 173 min, respectively, and those of the segmentectomy group were 195 and 140 min, respectively. Console operation was significantly (approximately 30 min) shorter in the segmentectomy group than in the lobectomy group (*p* < 0.001). Two more robotic and nonrobotic staplers (mean 8.2 ± 2.3) were used in the segmentectomy group than in the lobectomy group (6.6 ± 2.6; *p* = 0.003).

During surgery, no emergent or cool conversion to OTS occurred. Here, emergent conversion is mainly defined as a situation in which a major vessel injury occurs and urgent thoracotomy is required [20], whereas cool or calmer conversion refers to a situation that is not so urgent but continuing robotic surgery is inappropriate and open thoracotomy or VATS conversion is needed [2]. Two patients (1.8%, 2/114) needed unplanned conversions to regular confronting VATS. One patient underwent an urgent conversion for a moderate hemorrhage from a pulmonary artery branch during left upper lobectomy in the introduction phase (bleeding, 290 g). Another patient underwent a calmer conversion during an extended right S6+S10a segmentectomy, where staplers could not be inserted appropriately because of lung lacerations (bleeding, 440 g).

The median postoperative time of chest tube removal was 0 (0–7) days, and the duration of postoperative hospitalization was 3 (1–9) days. The durations of chest tube placement and hospitalization were not significantly different between the two groups. Regarding postoperative complications, one patient in the segmentectomy group, whose drain was removed on the same day of surgery, required chest tube reinsertion due to pulmonary collapse on day 2. In this patient, the tube was removed on postoperative day 5 and the patient was discharged home on day 8. One patient in the lobectomy group experienced prolonged air leak (>5 postoperative days); the leak subsequently resolved, the drain was removed on day 7, and the patient was discharged home on day 9. Another lobectomy patient whose drain was removed on postoperative day 1 required chest tube reinsertion on day 2 because of worsening subcutaneous emphysema; the drain was removed on postoperative day 4, and the patient was discharged home on day 7. Another patient in the lobectomy group developed an acute pyothorax from a wound infection during outpatient observation at 3 weeks postoperatively and urgent VATS pleurodesis was performed; thereafter, this patient recovered and was discharged home on postoperative day 7.

### 3.3. Prognostic Outcomes

The median duration of postoperative observation was 21 (2–40) months. Four patients had a follow-up time of <3 months. None of the patients experienced recurrence. One patient who underwent right lower lobectomy for pulmonary metastasis of glottic carcinoma with an uneventful postoperative course died unexpectedly 3 months postoperatively, probably from cardiovascular or cerebrovascular disease (details unknown). All other patients were well during the observation period.

## 4. Discussion

Surgical views are the most important aspect of this procedure. Because it is difficult to describe this point quantitatively, we have tried to demonstrate it as clearly and visually as possible using the videos and figures. In this robotic OTVA setting, regardless of the side to be operated on, the craniocaudal axis of the patient is aligned with the horizontal direction of the surgeon console monitor, and the cranial and caudal sides of the intrathorax are always displayed on the right and left sides of the surgeon console monitor screen, respectively. Therefore, this setting allows the operating and assisting surgeons to have the same surgical view and procedural consistency with our OTS and confronting VATS. Because surgical views and OTVA settings are different from those in the well-established worldwide conventional approach, some specific technical considerations are needed.

In this setting, assistant B plays the role of the fourth arm. Therefore, we are fully recognizing and standardizing the crossing movements and its patterns between the robotic instruments and the tools of assistant B to avoid the interference (Appendix A). Moreover, because the robotic ports and target structures are close in this approach, the surgeons may sometimes consider it difficult to view the lungs from a full distance, particularly when cutting the lung parenchyma. Thus, we consider that a stable pneumothorax environment by an insufflation system (AirSeal) is crucial. Therefore, with regard to the technical aspects of this procedure, especially in segmentectomy compared to lobectomy, more careful and gentle maneuvers in cooperation with the assistant should be taken when mobilizing the lung parenchyma.

As regards camera maneuverability, the ventral or dorsal side of the pulmonary hilum can become visible by switching the 30°-robotic camera up or down accordingly. During segmentectomy, these visibility characteristics can be well utilized to recognize the hilum of the segments and bronchovascular structures, as shown in the videos. Furthermore, under the Firefly viewing condition, the use of bronchoscope from the anesthesiologist’s side highlights the bronchial branches with a green luminescence, which makes it easier to identify the bronchial structures (Figure 2c and Figure 4c, and Appendix A).

Most of our segmentectomies were still relatively simple upper lobe procedures, and we have not yet had much experience with more complex segmentectomies of the lower basal segments. Our OTVA appears to be advantageous for upper lobe segmentectomy. On the contrary, the conventional look-up-view procedure may be more advantageous for a single-direction segmentectomy of the basal segments [24].

Our procedure is a hybrid method rather than a purely robotic one. Contrary to the trend toward future robotic surgeries performed by a single surgeon without assistants, our approach still requires two assistants. However, a human assistant playing as the fourth arm to complete a fine surgery is not necessarily disadvantageous. Having two assistants close to the patient also ensures safety in cases of emergency rollout and conversion to OTS or VATS [2]. An alternative, modified three-arm look-up-view procedure was reported [25].

As our robotic OTVA is different from the most commonly used, well-established conventional look-up-view approach, our method may be controversial and may present supporting and detracting perspectives: the widely used, four-arm, look-up-view method is undoubtedly considered the current mainstream approach worldwide including Japan. On the contrary, some robotic surgeons prefer OTVA-type procedures. Yamazaki et al. [26,27] and Funai et al. [28] described their own four-arm robotic approaches in which the patient’s craniocaudal direction can be viewed horizontally. Although these methods, including our own, are considered minor, this OTVA-type approach is gradually being recognized. Similarly, as VATS has several approaches including the look-up and CUD monitor methods, various approaches can be considered for RLRs.

Our experiences for robotic OTVA segmentectomies are still premature. Single-institution data also limit the generalization of the methodology and the findings. Further experience at multiple institutions is needed to generate more data supporting OTVA so that it can become alternative technique among those already used to tailor treatments to individual patients.

## 5. Conclusions

The three-arm robotic OTVA using vertical port placement and CUD monitor setting, which actualize natural thoracotomy views, can be an alternative for segmentectomy and lobectomy. This method has specific advantages and limitations; thus, these unique characteristics should be recognized thoroughly.

## Figures and Tables

**Figure 1 jpm-12-01771-f001:**
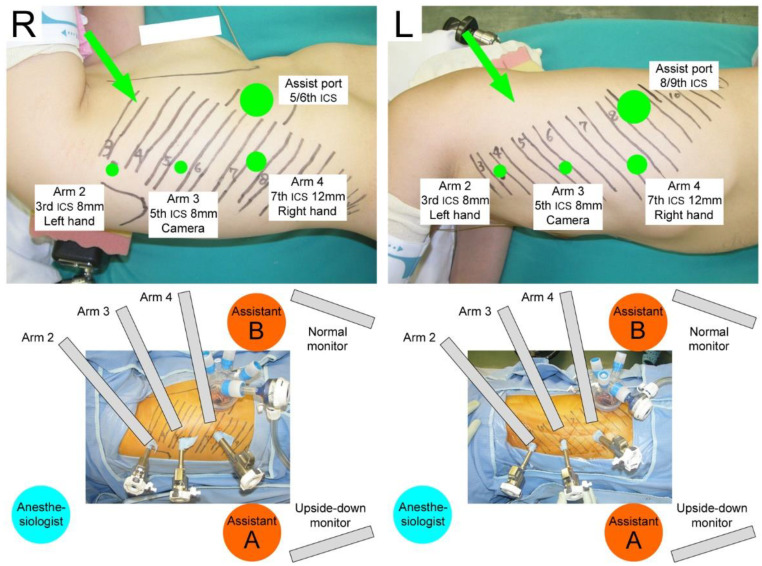
Vertical port placements (**top**) and settings of the robotic arms, two assistants, and confronting monitors (**bottom**) for right-side (**R**) and left-side (**L**) surgeries. The lines and numbers drawn on the patient’s body indicate the location of the ribs. The green circles indicate the incision size and intercostal space (ICS) where each port is placed. Arrows show the roll-in directions of the patient cart. These figures show the settings for the upper lobes. For middle and lower lobes, the port locations are caudally moved, as described in the text.

**Figure 2 jpm-12-01771-f002:**
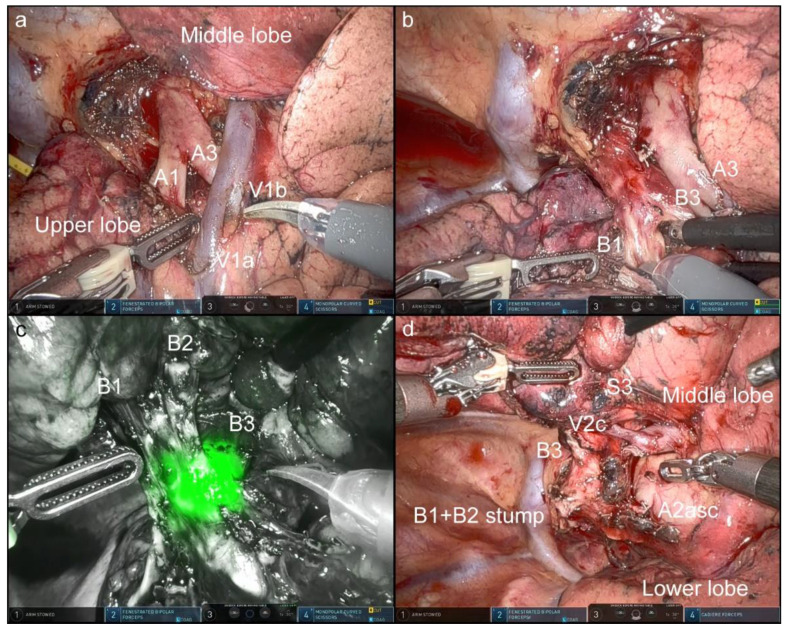
Right upper S1+2 segmentectomy via the three-arm, open-thoracotomy-view approach (Appendix A). The left and right sides of all images are the cranial and caudal sides of the intrathorax, respectively. Key images from the video are shown. Ventral side of the hilum (**a**). Segment-specific vessels (A1, V1a) were cut, and bronchi (B1) were dissected (**b**). The use of bronchoscope highlights the bronchial branches with a green luminescence under the Firefly viewing condition, which makes it easier to identify the bronchial structures (**c**). The completion of the procedure (**d**).

**Figure 3 jpm-12-01771-f003:**
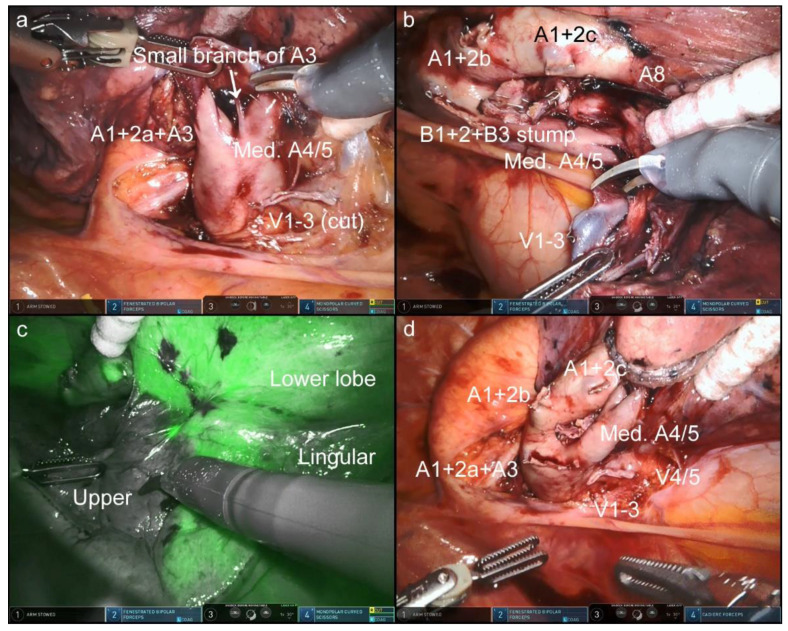
Left upper divisionectomy with mediastinal lingular pulmonary artery (Appendix A). At the ventral side of the hilum, a small branch of A3 was well confirmed (**a**). The mediastinal lingular pulmonary artery (A4/5) was fully elongated (**b**). Under the Firefly condition, segmental boundaries were clearly identified after the intravenous administration of indocyanine green (**c**). The completion of the procedure (**d**).

**Figure 4 jpm-12-01771-f004:**
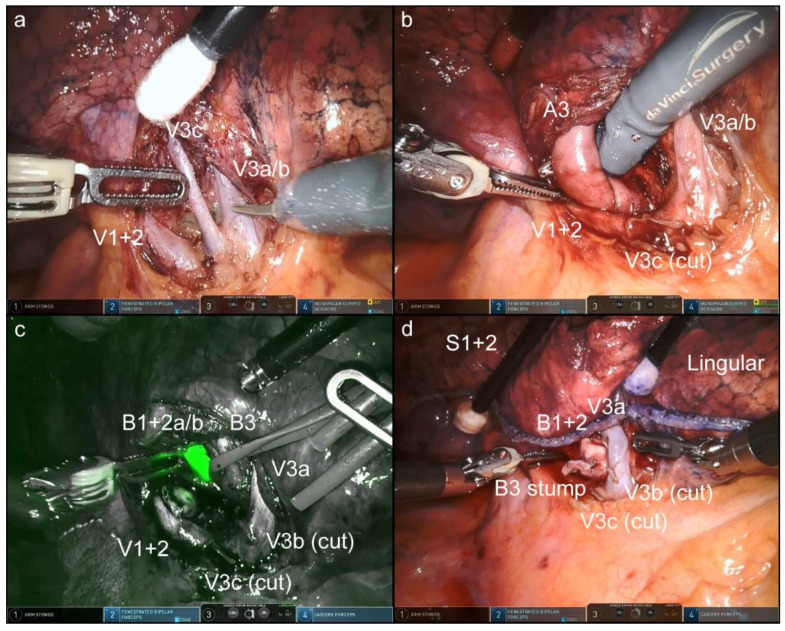
Left upper S3 segmentectomy (Appendix A). Segment-specific pulmonary vessel structures were well confirmed in the front (**a**,**b**). The use of bronchoscope highlights the bronchial branches with a green luminescence under the Firefly viewing condition, which makes it easier to identify the bronchial structures (**c**). The completion of the procedure (**d**).

**Figure 5 jpm-12-01771-f005:**
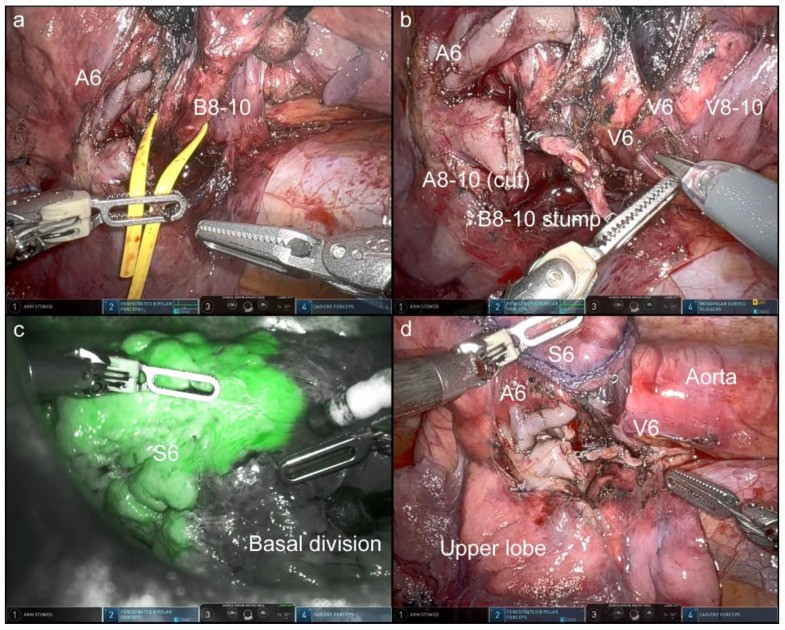
Left lower basal divisionectomy (Appendix A). After the basal pulmonary artery was cut, the basal bronchus was dissected (**a**). Dissection of pulmonary veins (**b**). Segmental boundaries were clearly identified after the intravenous administration of indocyanine green (**c**). The completion of the procedure (**d**).

**Table 1 jpm-12-01771-t001:** Baseline characteristics of the patients ^a^.

Variables		All Patients (*n* = 114)	Lobectomy (*n* = 76)	Segmentectomy (*n* = 38)	*p* Value ^b^
Age (median, range; years)		71 (36–86)	69 (36–86)	72 (56–86)	0.165
Sex					
	Male/female	45 (39)/69 (60)	30 (40)/46 (60)	15 (39)/23 (61)	1.000
Smoking status					
	Never/former or current	64 (56)/50 (44)	42 (55)/34 (45)	22 (58)/16 (42)	0.843
	Brinkman index (median, range)	0 (0–2040)	0 (0–1920)	0 (0–2040)	0.675
Body condition					
	Height (mean ± SD, range; cm)	160 ± 9 (140–181)	160 ± 9 (140–181)	158 ± 8 (145–178)	0.905
	Weight (mean ± SD, range; kg)	59 ± 11 (37–114)	59 ± 13 (37–114)	59 ± 9 (45–90)	0.579
	Body mass index (mean ± SD, range; kg/m^2^)	23 ± 3 (15–35)	23 ± 3 (15–35)	24 ± 3 (18–34)	0.162
Respiratory function					
	%VC (mean ± SD, range; % predicted)	102 ± 14 (62–152)	102 ± 13 (70–136)	101 ± 16 (62–152)	0.608
	%FEV1 (mean ± SD, range; % predicted)	99 ± 19 (40–172)	101 ± 16 (50–143)	96 ± 24 (40–172)	0.172
	%DLCO (mean ± SD, range; % predicted)	106 ± 23 (60–181)	108 ± 22 (69–181)	104 ± 25 (60–165)	0.340
HRCT findings and size					
	Pure GGO/partly solid/solid	11 (10)/69 (60)/34 (30)	6 (8)/45 (59)/25 (33)	5 (13)/24 (63)/9 (24)	0.469
	LD ^c^ (mean ± SD, range; cm)	2.1 ± 0.9 (0.7–5.7)	2.2 ± 0.9 (0.8–5.7)	1.9 ± 0.8 (0.7–5.0)	0.040
	CD ^c^ (mean ± SD, range; cm)	1.4 ± 0.8 (0–3.7)	1.5 ± 0.9 (0–3.7)	1.2 ± 0.7 (0–2.7)	0.065
	MD ^c^ (mean ± SD, range; cm)	0.8 ± 0.8 (0–3.5)	0.9 ± 0.9 (0–3.5)	0.5 ± 0.5 (0–2.1)	0.018
Preoperative diagnosis					
	Lung cancer (c-stage 0/IA1/IA2/IA3/IB)	110 (3/44/38/22/3)	73 (2/28/23/17/3)	37 (1/16/15/5/0)	0.347
	Metastatic lung tumor or other	4	3	1	

^a^ Data are presented as indicated or as the number of patients. ^b^ Lobectomy group vs. segmentectomy group. ^c^ LD, CD, MD were previously reported [21]. CD, consolidation dimension in HRCT lung window; DLCO, diffusing capacity of the lung for carbon monoxide; FEV1, forced expiratory volume in 1 s; GGO, ground-glass opacity; HRCT, high-resolution computed tomography; LD, whole tumor dimension in the HRCT lung window; MD, tumor dimension in HRCT mediastinal window; SD, standard deviation; VC, vital capacity.

**Table 2 jpm-12-01771-t002:** Surgical outcomes ^a^.

Variables					All Procedures (*n* = 114)	Lobectomy (*n* = 76)	Segmentectomy ^b^ (*n* = 38)	*p* Value ^c^
Operating time (mean ± SD, range; minutes)								
		Total time			215 ± 46 (128–368)	225 ± 44 (128–348)	195 ± 44 (138–368)	<0.001
		Console time			162 ± 43 (86–311)	173 ± 44 (89–311)	140 ± 32 (86–210)	<0.001
Surgical procedure								
		RU/RM/RL/LU/LL			41/11/26/24/12	36/11/15/7/7	5/0/11/17/5	
				RU S1/S1+2/S1+3/S3			1/2/1/1	
				RL S6/S8			9/1	
				LU S1+2+S3/S1+2/S3/S4+S5			7/8/2/0	
				LL S6/S8+9+10			3/1	
Node dissection								
		ND1/ND2a-1/ND2a-2			59 (52)/50 (44)/5 (4)	30 (39)/41 (54)/5 (7)	29 (76)/9 (24)/0 (0)	<0.001
Bleeding (median, range; g)								
					5 (1–440)	5 (1–290)	3 (1–440)	0.829
Number of staplers ^d^ (mean ± SD, range)								
					7.1 ± 2.5 (3–17)	6.6 ± 2.6 (3–16)	8.2 ± 2.3 (5–17)	0.003
Fibrin glue and polyglycolic acid sheet								
		–/+			36 (32)/78 (68)	24 (32)/52 (68)	11 (29)/27 (71)	0.832
Conversion								
		To VATS/to open			2/0	1/0	1/0	
Morbidity								
		Prolonged air leak (>5 days)			1	1	0	
		Subcutaneous emphysema			1	0	1	
		Chest tube reinsertion			2	1	1	
		Paroxysmal atrial fibrillation			1	1	0	
		Acute pyothorax			1	1	0	
Postoperative course (median, range; days)								
		Chest tube removal			0 (0–7)	0 (0–7)	0 (0–5)	0.289
		Hospital stay			3 (1–9)	3 (1–9)	3 (1–8)	0.430
Resection								
		R0/R1–2			114 (100)/0	76 (100)/0	38 (100)/0	
Histology								
		Primary lung cancer			107	71	36	
				Adenocarcinoma	100	66	34	
				Squamous cell carcinoma	5	4	1	
				Small cell carcinoma	1	1	0	
				Carcinoid	1	0	1	
			pT status					
				Tis/T1a/T1b/T1c/T2a/T2b/T3	4/33/45/15/9/0/1	2/17/33/13/5/0/1	2/16/12/2/4/0/0	0.076
			pN status					
				N0/N1/N2	105/2/0	69/2/0	36/0/0	0.148
			p-Stage					
				0/IA1/IA2/IA3/IB/IIA/IIB	4/33/45/14/8/0/3	2/17/33/12/4/0/3	2/16/12/2/4/0/0	0.049
		Metastatic lung tumor			4	2	2	
		Other			3	3	0	
Adjuvant chemotherapy								
	–/+				110/4	72/4	38/0	
Postoperative observation time (median, range; months)								
					21 (2–40)	23 (2–40)	16 (2–36)	
Prognosis								
		Local or distant recurrence			0	0	0	
		Dead/alive			1 ^e^/113	1 ^e^/75	0/38	

^a^ Data are presented as indicated or as the number of patients. ^b^ One patient who underwent right S6 segmentectomy and middle lobectomy for double primary lesions was included in the segmentectomy group because the primary lesion was located in the S6. ^c^ Lobectomy group vs. segmentectomy group. ^d^ Including robotic and nonrobotic staplers. ^e^ Because of cerebrocardiovascular disease of unknown details, described in the text. LL, left lower; LU, left upper; RL, right lower; RM, right middle; RU, right upper; SD, standard deviation; VATS, video-assisted thoracoscopic surgery.

## Data Availability

Data sharing is not applicable.

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
