# Peer review of "Three-Arm Robotic Lung Resection via the Open-Thoracotomy-View Approach Using Vertical Port Placement and Confronting Monitor Setting: Focusing on Segmentectomy"

_jpm, 2022, doi:10.3390/jpm12111771_

Round 1

Reviewer 1 Report

The description of the technique is quite complicated and extremely variable. It is a lot confusing, try to simplify it. the setup described in lines 97-100 is very tedious and difficult to be followed up. 

Line 73-74 contain a mistake: Segmentectomy group  vs segmentectomy group?

Line 79 "staples" is a writing mistake

Author Response

Response to Reviewer 1

We sincerely appreciate and thank you for highly valuable comments and constructive suggestions. We have tried to revise the manuscript in order to make the necessary corrections in line with the reviewers’ suggestions and comments as much as possible. All corrections in the revised manuscript are indicated in yellow highlight.

Comment 1: Reviewer inquired about details of the OTVA methodology:

The description of the technique is quite complicated and extremely variable. It is a lot confusing, try to simplify it. the setup described in lines 97-100 is very tedious and difficult to be followed up.

Answer 1: Thank you for your inquiry. Because our initial report on the 3-arm robotic OTVA method has already been reported in detail [1], some parts of it are omitted from this report, which may have made it difficult to understand the methodology. We would like to avoid reporting the same details as in the first report because it would be redundant. In this report, as per your suggestions, we describe a part of the methodology in detail, although it overlaps with the first report. The references of the initial and second reports are provided clearly [1,2].

Change 1: Line 91; lines 113-116

Comment 2: The reviewer pointed out a mistake in the description:

Lines 73-74 contain a mistake: Segmentectomy group vs segmentectomy group.

Answer 2: Thank you for your clarification. We have revised them correctly. The error has been revised correctly.

Change 2: Lines 170-172

Comment 3: The reviewer pointed out a mistake in the description:

Line 79, “staples”.

Answer 3: Thank you for your clarification. We have revised it correctly.

Change 3: Lines 172-173

Thank you again for your highly valuable comments and constructive suggestions. We hope that the manuscript is satisfactory. We would like to further investigate and refine our method, and express our sincerest gratitude for your kind consideration of our manuscript.

Reviewer 2 Report

In the Abstract:

—>  “focusing on segmentectomy and discuss its”: “focusing on segmentectomy and *discussing* its

—> “All procedures were performed safely”: I would avoid this sentence. One would assume that the OTVA was safe in your experience. Obviously, adverse events and bad outcomes can be described too and are a valuable source of information, but you would report this in an M&M conference setting or in a brief report. Moreover, one would hope that you would have stopped from using OTVA in the first place anyway, if all procedures were not performed safely.

If you want to keep this sentence to convey that 100% of your cases had good outcomes, you may say something like: “No adverse events resulted from the use of the OTVA technique”

—> “Cranial-side intrathoracic structures, which are often hidden in the conventional look-up-view method, were well visualized”: I think this is one of the major take away from your manuscript. Try to stress the positive throughout the paper, being honest about the limitations while still suggesting ways to turn such limitations into strenghts, moving forward.

—> “min”: I would avoid abbreviations here

—> “staplers were used compared with lobectomy (6.6 ± 2.6 staplers, p = 0.003)”: I would avoid repetitions like this one, *staplers* twice

—> “median postoperative durations of chest tube placement and hospitalization were 0 and 3 days, respectively”: 0 days meaning? Removed the same day of the surgery or not used at all? Also, what was the threshold for liquid loss? And did you apply an external source of suction or not?

—> “is an alternative for”: I would say '*can* be an alternative for'

In the Introduction:

—> “Alternatively, robotic lung segmentectomies have been widely performed [5-11]”: I am not sure I follow well here. What do you mean with “Alternatively”? There seem to be no logical connection with the previous paragraph.

—> “Herein, we describe our three-arm robotic OTVA”: you already did this in 2021 (reference 1). Please specify what you want to truly bring to the reader attention here, supporting the need for this new manuscript. 

In the Patients and Methods:

2.1. Patients

—> You talk a lot about these “145 patients” while 114 were the real focus of your paper. You did not carry out a prospective study where it makes more sense to describe thoroughly why patients were not included from the beginning or even more so why some of them fell out of the study.

Thus, my gut feeling is that you can simply state something like:

—> “Insurance coverage for RATS was initiated in April 2018 in Japan. In February 2019, we began to perform RATS at a full scale, totaling 145 surgeries until June 2022. Were retrospectively reviewed 114 consecutive major RLRs (lobectomy, n = 76; segmentectomy, n = 38) performed with the three-arm OTVA with the CUD monitor setting (Table 1). The institutional review board of the Aichi Cancer Center Hospital approved the study (#2021-0-223). Each patient provided informed consent for the use of their clinical data”.

To me, this is much more effective.

Table 1

and

2.2 Preoperative assessment and surgical indications

—> I wonder if “metastatic lung tumor or other” really fits in this paper. You’ve got a total of just 4 patients. Wouldn’t it just make more sense to remove them from the analysis?

I realize it may lead to more work, but it really does not add much to the manuscript anyway. Perhaps just visual clutter that requires more time for a careful reader to understand, without gaining substantial information rather than probably asking you: why have you performed a lobectomy for a metastatic lesion vs. local definitive treatment with RT +/- multi-modal treatment?

—> “three-dimensional reconstruction” I would specify the software you used for your 3D reconstructions

2.4. Segmentectomy and other surgical affairs

—> it sounds a bit odd to me to read “other surgical affairs”: either be more specific, or get rid of it

—> “In selecting Endowrist® instruments, we preferred using the combination of fenestrated bipolar forceps”: why so? I would take advantage of the opportunity to publish your technique to also include the reasons why you prefer something over something else

—> “For lymph node dissection, we frequently used Vessel Sealer Extend® and medium-large”: same here, why this over something else? Take this opportunity to share with us your feelings and/or reasons why you use one device/brand over others. We may be inspired and follow your lead if reasonable motivations are provided.

—> “Fibrin glue and polyglycolic acid”: please, provide the brand used

—> “20-Fr chest tube”: please, provide the type of tube

—> “according to the degree of air leak”: please, explain how you evaluated the degree of air leak

—> “If no air leak or chyle was detected “: please, provide the type of chest drainage system used to monitor your patient

—> “When no abnormalities were detected, the patients were discharged”: please, provide what would be abnormal in your clinical practice. Also, you can add *home* after discharged.

3. Results

3.1. Surgical views

—> “Surgical views are the most important aspect of this procedure”: to be deleted, this sentence does not belong to the result section.

—> for production team: I would make the Figure 1 pictures bigger to allow readers to appreciate the details and read the embedded text effortlessly. I would make the white text in Figures 2-5, 34 in bold, to allow readers to read it effortlessly.

3.2. Surgical outcomes and comparison between lobectomy and segmentectomy

—> “The baseline characteristics of the patients are shown in Table 1”: you already said this earlier. I would avoid repetitions unless strictly necessary or willingly done to make a point.

—> “The segmentectomy group tended to have smaller tumors and earlier disease stages than the lobectomy group, both in clinical and pathological stages”: I would be very careful here. I do not think you can say that the segmentectomy group showed smaller tumors because you selected this very criteria upfront - - - > 2.2. Preoperative assessment and surgical indications 

“As small lesions with an almost ground-glass appearance have a small pathological invasive size […] Segmentectomy was considered for such less invasive lesions, multiple lesions, or metastatic lung tumors”. It may be something to report in this result section data on pathological stages (although I wonder how you can say anything on the N status for segmentectomies if no mediastinal dissection was performed.

—> “All surgical procedures were performed safely”: I echo my prior comment

—> “Two more robotic and nonrobotic staplers were used in the segmentectomy group (mean 8.2 ± 2.3 staplers) than in the lobectomy group (mean 6.6 ± 2.6 staplers; p = 0.003)”: repetitions. I echo my prior comment.

—> “cool conversion”: sounds a bit funny to me, though I totally understand what you mean. Perhaps better using a more scientific term.

—> “Two patients (1.8%, 2/114) needed unplanned conversions to regular confronting VATS”: why?

—> “a calmer conversion ”: please, be more scientific

—> “(bleeding, 290 g) […] (blood loss amount, 440 g)”: if I were you, I would be consistent in the way I refer to blood loss. Choose one way to say it and keep it across.

—> “Most of the other surgeries were 180 performed with <10 g of blood loss”: the use of ‘most of’, in my opinion, is inappropriate. Either you provide numbers, or do not say it. Also, 10g blood loss is a very little amount to calculate, considering also an amount of water used intra-operatively. How did you quantify such 10grams?

—> “one patient in the segmentectomy group required chest tube reinsertion because of postoperative pulmonary collapse”: when was it removed in the first place?

—> “chest tube reinsertion because of worsening subcutaneous emphysema”: when was it removed in the first place?

—> “Another patient in the lobectomy group developed an acute pyothorax 3 weeks after surgery and subsequently underwent thoracoscopic pleurodesis; thereafter, this patient recovered quickly”: is it acute after 3 weeks? Do you do a pleurodesis for inflammatory fluid / pus in the pleural cavity? Lastly, again I would avoid the use of general terms like ‘quickly’: either provide data points or do not say it.

3.3. Prognostic outcomes 

—> “None of the patients experienced recurrence […] No local or distant recurrence occurred”: repetition. Avoid whenever possible.

In the Discussion:

—> “The three-arm robotic OTVA method has several limitations, as described previously [1,2] and herein. Our experiences for robotic segmentectomies are still premature, and further experience acquisition is needed. A retrospective analysis of data from a single institution also limits the generalization of the methodology and the findings”: I think this is a bit too harsh and you do not deserve it. You well explained pros and cons of your method earlier in the discussion section. You can simply state something like: “further research is needed to generate more data supporting OTVA so that it can became another technique among those already used by surgeons to tailor treatments to individual patients”.

—> “In conclusion, the three-arm OTVA using the vertical port placement and CUD monitor setting […]”: I would not keep this in the discussion. I would create a brief additional paragraph (5. Conclusions) and summarize your core conclusions. Stress the positive! It is important to capture the readers’ attention so that it goes back and read the manuscript in full.

Lastly:

—> Table formats: I would add horizontal lines to make them easier to read

—> More references are needed

Author Response

Response to Reviewer 2

We sincerely appreciate and thank you for highly valuable comments and constructive suggestions. We have tried to revise the manuscript in order to make the necessary corrections in line with the reviewers’ suggestions and comments as much as possible. All corrections in the revised manuscript are indicated in yellow highlight.

In the Abstract:

Comment 1: The reviewer pointed out the English description:

“focusing on segmentectomy and discuss its”: “focusing on segmentectomy and *discussing* its…

Answer 1: Thank you for the clarification. As per the suggestion, we have corrected it accordingly.

Change 1: Lines 28-29

Comment 2: The reviewer pointed out the English description:

“All procedures were performed safely”: I would avoid this sentence. One would assume that the OTVA was safe in your experience. Obviously, adverse events and bad outcomes can be described too and are a valuable source of information, but you would report this in an M&M conference setting or in a brief report. Moreover, one would hope that you would have stopped from using OTVA in the first place anyway, if all procedures were not performed safely. If you want to keep this sentence to convey that 100% of your cases had good outcomes, you may say something like: “No adverse events resulted from the use of the OTVA technique.”

Answer 2: Thank you for the clarification. As per the reviewer’s suggestion, we have deleted the sentence in the Abstract accordingly.

Change 2: Lines 33-35

Comment 3: The reviewer commented on understanding of the major take home message of this report:

“Cranial-side intrathoracic structures, which are often hidden in the conventional look-up-view method, were well visualized”: I think this is one of the major take away from your manuscript. Try to stress the positive throughout the paper, being honest about the limitations while still suggesting ways to turn such limitations into strenghts, moving forward.

Answer 3: Thank you for your comment. As you pointed out, the surgical view is the most important aspect of this approach. Because it is difficult to describe this point quantitatively, we have tried to demonstrate it as clearly and visually as possible using the videos. Accordingly, we have revised the relevant parts and have added this comment.

Change 3: Lines 159-160; lines 209-211

Comment 4: The reviewer pointed out the English description: “min”:

I would avoid abbreviations here

Answer 4: Thank you for the clarification. We have revised it accordingly.

Change 4: Line 38

Comment 5: The reviewer pointed out some repetitive descriptions:

—> “staplers were used compared with lobectomy (6.6 ± 2.6 staplers, p = 0.003)”: I would avoid repetitions like this one, *staplers* twice

Answer 5: Thank you for the clarification. We have corrected this accordingly.

Change 5: Lines 38-39

Comment 6: The reviewer suggested a more accurate description:

—> “median postoperative durations of chest tube placement and hospitalization were 0 and 3 days, respectively”: 0 days meaning? Removed the same day of the surgery or not used at all? Also, what was the threshold for liquid loss? And did you apply an external source of suction or not?

Answer 6: Thank you for the clarification. A chest drain was placed in all cases. The 0 day means that the drain was removed on the day of surgery. We have used a protocol of early postoperative mobilization [22,23]. We would like to use the term 0-day here. In the Methods section, we have explained this, citing the references. As per the suggestion, we have also added some information on drains to the Method section.

Change 6: Lines 143-149; lines 139-142

Comment 7: The reviewer suggested a revised statement in the Abstract’s conclusion:

—> “is an alternative for”: I would say '*can* be an alternative for'

Answer 7: Thank you for the suggestion. As per the suggestion, we have revised this sentence accordingly.

Change 7: Line 41

In the Introduction:

Comment 8: The reviewer pointed out the English description:

—> “Alternatively, robotic lung segmentectomies have been widely performed [5-11]”: I am not sure I follow well here. What do you mean with “Alternatively”? There seem to be no logical connection with the previous paragraph.

Answer 8: Thank you for the suggestion. We agree with the reviewer’s comment. We have revised this sentence.

Change 8: Lines 56-57

Comment 9: The reviewer suggested that the introduction should more clearly present the main points of the paper and the differences from previous report:

—> “Herein, we describe our three-arm robotic OTVA”: you already did this in 2021 (reference 1). Please specify what you want to truly bring to the reader attention here, supporting the need for this new manuscript.

Answer 9: Thank you for the clarification. We agree with the reviewer’s point of view. As per the suggestion, we have revised the bottom of the introduction.

Change 99: Lines 59-60

In the Patients and Methods:

Comment 10: The reviewer suggested that the description of the eligible patients should be more brief:

—> 2.1. Patients. You talk a lot about these “145 patients” while 114 were the real focus of your paper. You did not carry out a prospective study where it makes more sense to describe thoroughly why patients were not included from the beginning or even more so why some of them fell out of the study. Thus, my gut feeling is that you can simply state something like: “Insurance coverage for RATS was initiated in April 2018 in Japan. In February 2019, we began to perform RATS at a full scale, totaling 145 surgeries until June 2022. Were retrospectively reviewed 114 consecutive major RLRs (lobectomy, n = 76; segmentectomy, n = 38) performed with the three-arm OTVA with the CUD monitor setting (Table 1). The institutional review board of the Aichi Cancer Center Hospital approved the study (#2021-0-223). Each patient provided informed consent for the use of their clinical data”. To me, this is much more effective.

Answer 10: Thank you for your input. We agree with the reviewer’s view. We have revised the description brief, as suggested.

Change 10: Lines 64-70

Comment 11: The reviewer suggested excluding four cases in which the preoperative diagnosis was other than lung cancer:

Table 1—> I wonder if “metastatic lung tumor or other” really fits in this paper. You’ve got a total of just 4 patients. Wouldn’t it just make more sense to remove them from the analysis? I realize it may lead to more work, but it really does not add much to the manuscript anyway. Perhaps just visual clutter that requires more time for a careful reader to understand, without gaining substantial information rather than probably asking you: why have you performed a lobectomy for a metastatic lesion vs. local definitive treatment with RT +/- multi-modal treatment?

Answer 11: Thank you for the suggestion. Preoperative (Table 1) and postoperative (Table 2) diagnoses contain cases in which the diagnosis was changed. This was our clinical practice. Since our purpose of this paper is to evaluate consecutive patients who underwent anatomical lung resection using the 3-arm robotic OTVA, we would like to include cases other than lung cancer in the preoperative diagnosis, although the number of such cases is small. We have revised relevant parts. Please kindly allow us to include these cases in this report.

Change 11: Lines 73-75

Comment 12: The reviewer asked about tools for the three-dimensional construction of preoperative images:

—> “three-dimensional reconstruction” I would specify the software you used for your 3D reconstructions

Answer 12: Thank you for the inquiry. We have added this information in the revised manuscript.

Change 12: Lines 77-79

Comment 13: The reviewer pointed out the English description:

2.4. Segmentectomy and other surgical affairs —> it sounds a bit odd to me to read “other surgical affairs”: either be more specific, or get rid of it.

Answer 13: Thank you for the clarification. We followed the reviewer’s suggestion and revised the description.

Change 13: Line 129

Comment 14: The reviewer inquired about the reasons for the Endowrist selections:

—> “In selecting Endowrist® instruments, we preferred using the combination of fenestrated bipolar forceps”: why so? I would take advantage of the opportunity to publish your technique to also include the reasons why you prefer something over something else

Answer 14: Thank you for the inquiry. The reason for the preference for these instruments is that the primary surgeon (NS) prefers holding the long round point forceps in the left hand and the Metzenbaum scissors or Mayo scissors in the right hand during open thoracotomy surgery, and the monopolar curved scissors can cut tissue finely, although the dissecting speed is slower compared to the common Maryland bipolar forceps. We have added this comment in the revised manuscript.

Change 14: Lines 119-123

Comment 15: The reviewer inquired about the another reasons for the Endowrist selections during lymph node dissection:

—> “For lymph node dissection, we frequently used Vessel Sealer Extend® and medium-large”: same here, why this over something else? Take this opportunity to share with us your feelings and/or reasons why you use one device/brand over others. We may be inspired and follow your lead if reasonable motivations are provided.

Answer 15: Thank you for the inquiry. Currently, there are two sealing devices that can be utilized with the da Vinci Xi system: the Vessel Sealer Extended and the SynchroSeal. Because the latter is not yet available at our institution, we use the former for our robotic sealing maneuvers. We have added this information in the revised manuscript.

Change 15: Line 125

Comment 16: The reviewer asked for manufacturers of fibrin glue and PGA sheet:

—> “Fibrin glue and polyglycolic acid”: please, provide the brand used

Answer 16: Thank you for the inquiry. We have added this information in the revised manuscript.

Change 16: Lines 137-138

Comment 17: The reviewer asked for more information about thoracic drains:

—> “20-Fr chest tube”: please, provide the type of tube

Answer 17: Thank you for the inquiry. We have added this information.

Change 17: Line139

Comment 18: The reviewer inquired about details on intraoperative air leak assessment and sealing test:

—> “according to the degree of air leak”: please, explain how you evaluated the degree of air leak

Answer 18: Thank you for the inquiry. We have added this information in the revised manuscript, accordingly.

Change 18: Lines 135-136

Comment 19: The reviewer inquired about postoperative chest drainage system:

—> “If no air leak or chyle was detected “: please, provide the type of chest drainage system used to monitor your patient

Answer 19: Thank you for the inquiry. We have added this information in the revised manuscript.

Change 19: Lines 140-143

Comment 20: The reviewer asked for postoperative management:

—> “When no abnormalities were detected, the patients were discharged”: please, provide what would be abnormal in your clinical practice. Also, you can add *home* after discharged.

Answer 20: Thank you for the inquiry and input. We adopted and reported a protocol for early postoperative mobilization with drain removal on the same day of surgery [22,23]. We have added this information and revised the relevant parts accordingly.

Change 20: Lines 143-149

In the Results:

Comment 21: The reviewer pointed out that the comment was improperly positioned:

3.1. Surgical views, —> “Surgical views are the most important aspect of this procedure”: to be deleted, this sentence does not belong to the result section.

Answer 21: Thank you for the clarification. We agree with the reviewer’s opinion. The sentence was deleted, and the relevant parts were revised accordingly. We have also revised the beginning of our discussion using this message.

Change 21: Lines 209-211

Comment 22: The reviewer ordered the editorial team to adjust the size of Figures.

—> for production team: I would make the Figure 1 pictures bigger to allow readers to appreciate the details and read the embedded text effortlessly. I would make the white text in Figures 2-5, 34 in bold, to allow readers to read it effortlessly.

Answer 22 Thank you for comments. We are sorry, but we cannot confirm the size of the figure for peer review and therefore cannot comment on it. In this paper, the videos and figures that visually show the surgical view are most important. We respect your advice, we will coordinate with the editorial team regarding the figures.

Change 22: Contents of figures 1-5 retained

Comment 23: The reviewer suggested removing repetitive phrases:

3.2. Surgical outcomes and comparison between lobectomy and segmentectomy

—> “The baseline characteristics of the patients are shown in Table 1”: you already said this earlier. I would avoid repetitions unless strictly necessary or willingly done to make a point.

Answer 23: Thank you for the clarification. We agree with the reviewer’s opinion. As per the suggestion, this was deleted, and the relevant parts were revised.

Change 23: Line 167

Comment 24: Reviewer inquired about pathological staging. And also, reviewer asked about the N factor in cases without mediastinal dissection.

—> “The segmentectomy group tended to have smaller tumors and earlier disease stages than the lobectomy group, both in clinical and pathological stages”: I would be very careful here. I do not think you can say that the segmentectomy group showed smaller tumors because you selected this very criteria upfront - - - > 2.2. Preoperative assessment and surgical indications

“As small lesions with an almost ground-glass appearance have a small pathological invasive size […] Segmentectomy was considered for such less invasive lesions, multiple lesions, or metastatic lung tumors”. It may be something to report in this result section data on pathological stages (although I wonder how you can say anything on the N status for segmentectomies if no mediastinal dissection was performed.

Answer 24: Thank you for the inquiries and clarifications. We agree with the reviewer’s opinion and also consider the descriptions to be inadequate. We have correctly revised the relevant parts. In cases where mediastinal lymph node dissection was omitted, possible hilar lymph nodes were evaluated and their pN factor and p-Stage were determined. We have also added this information in the Methods section.

Change 24: Lines 167-168; lines 86-88

Comment 25: The reviewer suggested that the unsuitable statement be deleted:

—> “All surgical procedures were performed safely”: I echo my prior comment

Answer 25: Thank you for the suggestion. As per the suggestion, the sentence was deleted.

Change 25: Deleted

Comment 26: The reviewer suggested avoiding repetitive expressions “staplers”:

—> “Two more robotic and nonrobotic staplers were used in the segmentectomy group (mean 8.2 ± 2.3 staplers) than in the lobectomy group (mean 6.6 ± 2.6 staplers; p = 0.003)”: repetitions. I echo my prior comment.

Answer 26: Thank you for the clarification. We agree and have corrected relevant part accordingly.

Change 26: Lines 172-173

Comment 27: The reviewers suggested a more detailed description for cool and calmer conversions:

—> “cool conversion”: sounds a bit funny to me, though I totally understand what you mean. Perhaps better using a more scientific term.

—> “a calmer conversion ”: please, be more scientific

Answer 27: Thank you for the clarification. You fully know the clinical meaning of cool conversion. Here, emergent conversion is mainly defined as a situation in which a major vessel injury occurs and urgent thoracotomy is required [20], whereas cool or calmer conversion refers to a situation that is not so urgent but continuing robotic surgery is inappropriate and open thoracotomy or VATS conversion is needed [2]. We have added these explanations to complement the current description.

Change 27: Lines 174-178

Comment 28: The reviewer inquired about the reasons for the two unexpected VATS conversion cases:

—> “Two patients (1.8%, 2/114) needed unplanned conversions to regular confronting VATS”: why?

 Answer 28: Thank you for the inquiry. The reasons for these cases were described subsequently.

Change 28: Lines 179-183

Comment 29: The reviewers suggested that the different descriptions of bleeding volume be standardized:

—> “(bleeding, 290 g) […] (blood loss amount, 440 g)”: if I were you, I would be consistent in the way I refer to blood loss. Choose one way to say it and keep it across.

Answer 29: Thank you for pointing this out. We have revised this description and made it consistent with the Table 2.

Change 29: Lines 179-183

Comment 30: The reviewer inquired about the bleeding assessment:

—> “Most of the other surgeries were 180 performed with <10 g of blood loss”: the use of ‘most of’, in my opinion, is inappropriate. Either you provide numbers, or do not say it. Also, 10g blood loss is a very little amount to calculate, considering also an amount of water used intra-operatively. How did you quantify such 10grams?

Answer 30: Thank you for pointing that out. The description was ambiguous and inappropriate. During our surgery, bleeding was suctioned by the assistant and measured, and our robotic and VATS surgeries cause very little bleeding, so the blood loss was very small, as shown in Table 2. Here, as per the suggestion, we have removed the description and revised relevant parts.

Change 30: Lines 179-183

Comment 31: The reviewer inquired about the details of the cases in which the drains were reinserted:

—> “one patient in the segmentectomy group required chest tube reinsertion because of postoperative pulmonary collapse”: when was it removed in the first place?

—> “chest tube reinsertion because of worsening subcutaneous emphysema”: when was it removed in the first place?

Answer 31: Thank you for the inquiries. We have added more information about these cases.

Change 31: Lines 187-190; lines 192-195

Comment 32: The reviewer inquired about the postoperative pyothorax case:

—> “Another patient in the lobectomy group developed an acute pyothorax 3 weeks after surgery and subsequently underwent thoracoscopic pleurodesis; thereafter, this patient recovered quickly”: is it acute after 3 weeks? Do you do a pleurodesis for inflammatory fluid / pus in the pleural cavity? Lastly, again I would avoid the use of general terms like ‘quickly’: either provide data points or do not say it.

Answer 32: Thank you for your inquiry and the input. We have revised this and added more information about the patient.

Change 32: Lines 195-198

Comment 33: The reviewer suggested avoiding repetitive expressions:

3.3. Prognostic outcomes —> “None of the patients experienced recurrence […] No local or distant recurrence occurred”: repetition. Avoid whenever possible.

Answer 33: Thank you for the clarification. We have deleted one of the repetitive expressions and have corrected relevant part.

Change 33: Line 202

In the Discussion:

Comment 34: The reviewer suggested a revision to the description of limitation.

—> “The three-arm robotic OTVA method has several limitations, as described previously [1,2] and herein. Our experiences for robotic segmentectomies are still premature, and further experience acquisition is needed. A retrospective analysis of data from a single institution also limits the generalization of the methodology and the findings”: I think this is a bit too harsh and you do not deserve it. You well explained pros and cons of your method earlier in the discussion section. You can simply state something like: “further research is needed to generate more data supporting OTVA so that it can became another technique among those already used by surgeons to tailor treatments to individual patients”.

Answer 34: Thank you for your input. Since this OTVA method differs from the common method of robotic lung resections, we would like to describe some of the points. We have revised the limitation brief based on the suggested statements.

Change 34: Lines 257-260

Comment 35: The reviewer advised that the conclusion should be in a separate section to emphasize the take home message in a clear manner.

—> “In conclusion, the three-arm OTVA using the vertical port placement and CUD monitor setting […]”: I would not keep this in the discussion. I would create a brief additional paragraph (5. Conclusions) and summarize your core conclusions. Stress the positive! It is important to capture the readers’ attention so that it goes back and read the manuscript in full.

Answer 35: Thank you for your suggestion. In accordance with the your input, we created a paragraph “5. Conclusion” to emphasize our message.

Change 35: Lines 262-266

Comment 36: The reviewer mentioned tables and references.

Lastly:

—> Table formats: I would add horizontal lines to make them easier to read

—> More references are needed

Answer 36: Thank you for your suggestion. We have added horizontal lines in the tables. These may be modified according to the journal’s table formatting rules. In addition, we have added some articles to the references.

Change 36: Horizontal lines in Tables 1 and 2; references [12,13,15,20,22,23].

Thank you again for your highly valuable comments and many constructive suggestions. We hope that the manuscript is satisfactory. We would like to further investigate and refine our method, and express our sincerest gratitude for your kind consideration of our manuscript.

Reviewer 3 Report

Dear Authors,

I read this manuscript with interest as I did with your previous report on the same topic. I do not think the reported data will substantially change the fashion of RAST surgery techniques in treating early NSCLC. Anyway, the manuscript is well designed, and the methods are precisely described and followed by excellent figures and videos. 

I perceived a bit of redundancy considering the paper you already published, but the discussion provides sufficient justification and argumentation for those who could read this manuscript at first.

If possible, I only suggest avoiding the reporting results in the text when they are already mentioned in the table.

Compliments.

Author Response

Response to Reviewer 3

We sincerely appreciate and thank you for highly valuable comments and constructive suggestions. We have tried to revise the manuscript in order to make the necessary corrections in line with the reviewers’ suggestions and comments as much as possible. All corrections in the revised manuscript are indicated in yellow highlight.

Comment 1: The reviewer suggested avoiding the reporting results in the text when they are already mentioned in the table:

If possible, I only suggest avoiding the reporting results in the text when they are already mentioned in the table.

Compliments.

Answer 1: Thank you for the clarification. As per the suggestion, we have removed some of the repeated statements that are also described in the tables. Some of the statements we would like to emphasize were revised and retained.

Change 1: Lines 167-173

Thank you again for your highly valuable comments and constructive suggestions. We hope that the manuscript is satisfactory. We would like to further investigate and refine our method, and express our sincerest gratitude for your kind consideration of our manuscript.

Round 2

Reviewer 1 Report

In my opinion the description is too much chaotic and auto-reffered. Does this technique add something to the clinical practice?

Author Response

Response to Reviewer 1

We sincerely appreciate and thank you for highly valuable comments and suggestions. In this second peer review, reviewer #1 carefully read our inadequate descriptions and pointed them out. We have tried to revise the manuscript in order to make the necessary corrections in line with the reviewer’s suggestions and comments as much as possible.

Comment: The reviewer pointed out that the description of the OTVA methodology was inadequate, and the reviewer inquired about the possible benefits of the method.

Answer: Thank you again for pointing out. Since our initial report on the 3-arm robotic OTVA method has already been reported in detail using videos [1], we omitted some descriptions of the method in this previous paper, which made the method incomprehensible and insufficient. There were also some errors in the description of port placement. We apologize for these points.

We have revised the relevant section (2.3. Three-arm robotic OTVA setting) extensively by dividing it into separate subsections to make it more comprehensible. In addition, we have added descriptions and explanations to complement understanding the methodology in consistent with the Figure 1.

This OTVA method allows the operating and assisting surgeons to have the same surgical view in our open thoracotomy surgery and VATS. To maintain this consistency even in robotic lung resections, we devised our three-arm robotic OTVA, as described in the introduction (line 46-52), results (lines 169-175) and discussion (lines 220-227) sections. Because surgical views and OTVA settings are different from those in the well-established worldwide conventional look-up-view approach, some considerations are needed for this method. In this paper, we would like to present the technical aspects of the method focusing on segmentectomy.

Revisions suggested by each reviewer in the first review are retained in yellow highlight. Further revisions made in this second review are shown in green highlight.

Please kindly allow us to revise our report as described.

Change: 2.3. Three-arm robotic OTVA setting section; lines 86-138

Thank you again for your careful review. We would like to further investigate and refine our method, and express our sincerest gratitude for your kind consideration of our manuscript.